# Late adolescents' own and assumed parental preferences towards health-care related confidentiality and consent in Belgium

**David De Coninck**[1]*, **Koen Matthijs**[1], **Peter de Winter**[2,3], **Jaan Toelen**[3,4]

1 Centre for Sociological Research, KU Leuven, Leuven, Belgium, 2 Department of Pediatrics, Spaarne Gasthuis, Hoofddorp, The Netherlands, 3 Department of Development and Regeneration, KU Leuven, Leuven, Belgium, 4 Division of Woman and Child, Department of Pediatrics, UZ Leuven, Leuven, Belgium

* david.deconinck@kuleuven.be

## Abstract

### Objectives

Health care professionals regularly struggle with issues relating to confidentiality and consent for physical and/or mental health issues among adolescents. We investigate late adolescents' own and assumed parental preferences towards health-care related confidentiality and consent.

### Methods

We analyzed online survey data of four vignettes from 463 first-year university students at KU Leuven (Flanders, Belgium). We used paired samples t-tests to assess the (in)consistency between attitudes of late adolescents and their assumed parental attitudes, independent samples t-tests to estimate gender differences, and binomial logistic regressions to analyze the association of assumed parental preferences with late adolescents' own preferences.

### Results

Attitudinal inconsistencies were present in all vignettes. Late adolescents were significantly more in favor of confidentiality and adolescent consent than what they believed their parents were. Gender differences were limited. Binomial logistic regressions indicated that assumed parental preferences were strongly associated with late adolescents' own preferences.

### Conclusions

Findings suggest a clear difference between late adolescents' preferences and assumed parental preferences: they believe that their parents are less inclined to favor confidentiality and adolescent consent. We also find that this difference depends on the case, indicating that there is no such thing as general 'confidentiality preferences'. Rather, a decision- and/ or context-specific perspective should be adopted.

**Data Availability Statement:** Data cannot be shared publicly because of ethical concerns. However, data are available on request for researchers who meet the criteria for access to

confidential data. Requests may be sent to the corresponding author (david.deconinck@kuleuven.be) and the Social and Societal Ethics Committee of KU Leuven (smec@kuleuven.be).

**Funding:** The author(s) received no specific funding for this work.

**Competing interests:** The authors have declared that no competing interests exist.

## Background

Health care professionals regularly struggle with providing care to adolescents, especially in relation to issues of confidentiality and consent in terms of physical and/or mental health issues [1]. Although many adolescents have the maturity to decide the course of their health-related decisions, their right to confidentiality sometimes rests in the hands of those professionals who provide their care [2]. Over the past 30 years, the consideration of underage patients' role in decision-making has evolved [3]. Many European countries implemented regulations that allow confidential services for minors of a certain age, or minors who possess adequate decision-making capacity [4]. Generally, sexual and reproductive health conditions often do not require consent from a parent or guardian [2]. In Europe, seven countries define the age at which a child is competent to consent to medical interventions or clinical research based on 'maturity' (i.e. are sufficiently mature and understand the nature and implications of the treatment; for instance the 'Gillick competence' in the UK), 11 countries allow this from the age of 16 years and ten countries from the age of 18 years [1–3, 5–8]. In short, there are a variety of consent systems at play in Europe. If adolescents under 16 years of age have enough decision-making capabilities, they can become the principal decision maker for themselves in specific cases [5–7]. This is also reflected in article 12 of the Belgian Law on Patient's Rights (22nd August 2002) that states that a patient, regardless of age and level of development, should be involved in exercising their rights. If they are capable of exercising these rights, they can do so without mediation or intervention by parents or guardians [9]. However, just because minors are legally allowed make their own decisions, does not necessarily mean that they will do so. For instance, they may not be aware of their rights as patients [10], but even if they are, the reality of this is often more complicated than its legal framework makes it seem. While health-care related decisions should be free of biases like false assumptions, misinformation, and external pressure, the reality is that late adolescents do not make such decisions in a social vacuum.

In particular, parents' views may affect the way minors view their own health care, especially when parents believe that their child should not be allowed to make their own decisions below a certain age, regardless of legal considerations [3, 6]. Parental attitudes towards health-care related confidentiality and consent for adolescents are mixed. Although parents recognize the benefits associated with confidentiality for their children, they also view such health-care related information as a parental right and are uncomfortable not knowing what is discussed in these confidential consultations [11–13]. Although only very few studies have been conducted that investigate parents' reasoning behind these mixed feelings, preliminary evidence from Australia indicates that these feelings are underpinned by two key factors: the way in which they perceived their role as a parent and their level of trust in health professionals generally, but specifically, their child's physician [12]. European evidence on this topic is scarce [13].

However, there is a paucity of data regarding physicians' respect for confidentiality in adolescent healthcare in Europe. Studies from Lithuania, Belgium and Spain show that physicians are reluctant to spend time alone with adolescents, and that they often inform parents without asking adolescents for permission [4, 8, 14, 15]. Despite this, physicians' attitudes are often a determining factor in their final decision to protect adolescents' confidentiality, despite legal policy that promotes adolescents' rights [4, 5]. Recent studies have shown that adolescents identify a lack of confidentially as a barrier to seeking health care. Instead, they prefer to seek this care and communicate with physicians who assure confidentiality [16–19]. Furthermore, adolescents may choose to forgo health care to avoid the risk of their parents finding out about it [17, 20–23]. This is problematic as "access to health care is especially important during adolescence because it may modify risky behaviors, promote healthy habits, and improve health"

[18, 19, 24]. Even though guidelines indicate that physicians should excuse parents during the medical examination of adolescents in order to increase the likelihood that adolescents will share sensitive information, many fail to do so [14, 25–27].

Jackson et al. [2] state that the issue of confidentiality leads into other areas of interest as well, including whether or not an adolescent can refuse or consent to their own treatment without or against the consent of their parent(s) or guardian(s)", or in short, whether they are 'competent' to consent to medical interventions or clinical research [2]. There are several circumstances in which this may become a problem for health professionals (e.g. a minor requesting an abortion or refusing to receive blood products due to religious beliefs). Making a decision in such cases is not simply deciding whether to provide treatment or not. Although efforts have been made to improve training in adolescent health care, studies have shown that a high proportion of physicians still "feel uncomfortable with providing services for medically emancipated conditions [i.e. sensitive health issues, like sexual or reproductive health] and/or providing confidential care" to adolescents without informing or asking consent of their parent or guardian [3, 20, 28–32], regardless of the system in place. Studies in several countries (including Belgium, the setting of this study) have shown that less than 50% of patients (of all ages) are aware of their rights, and that these rights are not always respected [10, 33].

In some contexts, the participants of this study (first-year university students) could be described as 'young adults' rather than 'late adolescents'. However, given this group's narrow age range, a sizeable share (39%) of our sample is under 18 years old which–given the Belgian consent system–could mean that they are not considered medically competent and can therefore not consent to medical interventions or clinical research without a parent or guardian. The rest of the sample only recently passed this legal threshold. On top of this, recent neuroimaging studies have shown that the adolescent brain continues to mature well into the early 20s or later [34, 35], while UNICEF emphasizes that late adolescence encompasses the latter part of the teenage years (usually between 15 to 19 years) [36]. This indicates that 'late adolescents' is, from both a biological and legal-medical perspective, a more accurate description of the study participants than 'young adults'.

In this study, we aimed to investigate the degree to which late adolescents' own perceptions of health-care related confidentiality and consent differed from what they believe their parents thought about this. In order to do so, we conducted an online survey among a sample of first-year university students at KU Leuven (Belgium) to assess confidentiality and consent perceptions among late adolescents. Although previous studies compared adolescents' attitudes on confidentiality to those of their parents [27, 37], our study built on this literature by assessing late adolescents' assumptions of their own parents' attitudes. Thus, this study also investigated assumed parental perceptions rather than parents´ actual perceptions. This is potentially even more worthwhile to investigate since late adolescents' own beliefs about their parents' confidentiality and consent preferences are equally likely to impact their behavior rather than their parents' actual beliefs, which are sometimes unknown [7].

## Materials and methods

### Data collection

We collected data through an online questionnaire among a convenience sample of students in a sociology course at the Faculty of Psychology and Educational Sciences at KU Leuven (Flanders, Belgium) in 2019. The assessment took place at the start of the semester, during the first class at the end of September. The questionnaire was programmed in Qualtrics, an online platform for developing questionnaires and collecting data. A URL linking to the questionnaire was made available on the official student portal a few minutes before the class. Students

were asked to participate in the study during class via smartphone, tablet, or laptop. If this was not possible, they had the opportunity to complete the questionnaire at home the same day. 463 students completed the assessment (response rate: 84%). The survey language was Dutch, the language of instruction [38, 39]. This study was approved by the Social and Societal Ethics Committee of KU Leuven (G-2017 09 934), and written informed consent was obtained from each respondent. Because all underage participants were 17 years old and active students at this university, we were not required to obtain informed consent from their parents or guardians. All methods were carried out in accordance with relevant guidelines and regulations for experiments where humans were involved and/or human data was collected.

## Measures

To gauge respondents' own and assumed parental preferences towards confidentiality and consent, we presented them with four fictional vignettes (cases). Each case concerned a situation where an unidentified 15-year old adolescent has received or needs to receive medical care which they either do not want to disclose to their parents or receive without their parents' consent. For each case, the respondent was asked to indicate their own preference regarding confidentiality or consent, and which option that they believed their own parents would choose. We set the age of the fictitious adolescent at 15 years old because this is an age at which–in 18 out of 28 EU countries, including Belgium–adolescents are not yet considered to be competent 'by default' but must rather be found competent by a physician [5].

Case one concerned a 15-year old adolescent who consumed alcohol with friends during a night out, falls over, cuts his hand on a piece of glass and is taken to the emergency room (shortened to 'drunk' hereafter). Case two concerned a 15-year old who was informed that he/ she requires oral and maxillofacial surgery to fix an underbite. This underbite has been the source of teasing, but the parents feel that surgery is dangerous and redundant (shortened to 'surgery'). In case three, a 15-year old teenager was diagnosed with a sexually transmitted disease that could be treated by a course of antibiotics for him/her and his/her partner (shortened to 'std'). In the fourth case, a 15-year old female visited her physician following several months of severe menstrual pains. The physician suggests that she starts taking birth control to manage these pains. However, her parents are not in favor, and do not agree with this treatment (shortened to 'pill'). This case resembles the one that originally spurred the lawsuit which resulted in the development of the Gillick competence, in which a mother of girls under 16 years old objected to the Department of Health advice that allowed doctors to give contraceptive advice and treatment to children without parental consent [6]. Respondents had to indicate whether they were of the opinion that the physician should inform the parents (confidentiality; cases one and three) or proceed with treatment without the parents' consent (consent; cases two and four) (0 = no, 1 = yes). Directly after this assessment, respondents were asked to indicate what they believed their own parents would prefer (0 = no, 1 = yes). In order to interpret mean scores and regression coefficients more easily, we reverse coded the scores for case one and three so that a higher score corresponded to greater support for confidentiality among adolescents and their parents. The full vignettes can be found in S1 Appendix, and a descriptive overview of no/yes-answers in S1 Table.

Respondents were also asked to indicate age, gender (1 = male, 2 = female), type of secondary education (1 = general secondary education, 2 = vocational secondary education, 3 = artistic secondary education, 4 = technical secondary education), mothers' educational attainment (1 = primary or secondary education, 2 = tertiary education), and parental cohabitation status (1 = intact (married or unmarried cohabitation), 2 = non-intact (legally divorced, separated, never cohabited)).

## Data analysis

We conducted a Pearson correlation analysis to establish the extent to which the vignette scores of respondents' own preferences and the assumed parental preferences correlated. The strongest Pearson coefficients were found between vignettes within each type (confidentiality and consent). We investigated differences within and between vignette scores by using paired samples t-tests to compare scores of respondents' own perceptions to those from the assumed parental preferences, after which we investigated gender differences within each vignette through independent samples t-tests. Finally, we conducted binomial logistic regressions to investigate the association between assumed parents' perceptions and respondent's own perceptions for each case. All analyses were conducted in SPSS version 25.

# Results

## Descriptive overview of the sample and study variables

Our sample consisted mainly of girls, which was common for the student population in this faculty. Most respondents had completed general secondary education and had highly educated mothers who were mostly married or cohabiting. Of the total sample, over 93% was between 17 and 19 years old (age range = 17 to 23; mean age = 18.11 years, $SD$ = 0.96). An overview of the sample is presented in Table 1. The results in Table 2 indicate that only weak to moderate correlations existed within and between vignettes.

## Differences in confidentiality and consent preferences

The results in Table 3 indicate that respondents held different preferences towards confidentiality and consent than what they believed their parents held. Mean scores indicated that with regard to

**Table 1. Sociodemographic distribution of sample.**

|  | N | % |
|---|---|---|
| **Age** | | |
| 17 years old | 98 | 21.2 |
| 18 years old | 269 | 58.1 |
| 19 years old | 67 | 14.5 |
| 20 to 23 years old | 29 | 6.2 |
| **Gender** | | |
| Female | 389 | 84.0 |
| Male | 74 | 16.0 |
| **Type of secondary education** | | |
| General secondary education | 414 | 89.4 |
| Technical secondary education | 37 | 8.0 |
| Artistic secondary education | 6 | 1.3 |
| Vocational secondary education | 3 | 0.6 |
| *Missing* | *3* | *0.6* |
| **Mothers' educational attainment** | | |
| Primary or secondary education | 114 | 24.7 |
| Tertiary education | 324 | 70.0 |
| *Missing* | *25* | *5.3* |
| **Parental cohabitation status** | | |
| Intact | 297 | 64.2 |
| Non-intact | 129 | 27.8 |
| *Missing* | *37* | *8.0* |

**Table 2. Pearson correlation coefficients between vignette scores.**

| | | Confidentiality | | | | Consent | | | |
|---|---|---|---|---|---|---|---|---|---|
| | | Drunk$_a$ | Drunk$_p$ | Std$_a$ | Std$_p$ | Surgery$_a$ | Surgery$_p$ | Pill$_a$ | Pill$_p$ |
| Confidentiality | Drunk$_a$ | 1 | | | | | | | |
| | Drunk$_p$ | .25** | 1 | | | | | | |
| | Std$_a$ | .31** | .01 | 1 | | | | | |
| | Std$_p$ | .13** | .21** | .38** | 1 | | | | |
| Consent | Surgery$_a$ | .04 | .01 | -.03 | -.02 | 1 | | | |
| | Surgery$_p$ | .01 | -.09* | -.03 | -.10 | .47** | 1 | | |
| | Pill$_a$ | .00 | .14** | -.11** | -.03 | .18** | .09* | 1 | |
| | Pill$_p$ | .02 | .03 | .03 | -.14** | .10* | .22** | .39** | 1 |

Note: The subscript a refers to respondents' own preferences. The subscript p refers to assumed parental preferences.

*: p < 0.05;

**: p < 0.01. Drunk = case 1, Surgery = case 2, Std = case 3, Pill = case 4.

the cases on drunkenness and the sexually transmitted disease, which both frame issues of confidentiality, respondents expected that their parents wanted to be informed, while respondents themselves would prefer the health care professional to respect the confidentiality. However, there was some variation between cases: respondents felt more strongly that the physician should remain confidential in the case of the sexually transmitted disease than for the drunkenness. As for the cases concerning jaw surgery and contraceptive use, both framing issues of treatment without the parents' consent, respondents felt that the physician should follow the adolescent's wishes. Although respondents believed their parents generally shared this preference for the case on maxillofacial surgery, this was less so for the case on contraceptive use: a majority of respondents believed their parents would not want the treatment to go ahead without their consent.

When we investigated gender differences within vignettes, we found that there was one statistically significant difference between boys and girls (Table 4): girls were found to be significantly more in favor of confidentiality in the case of the maxillofacial surgery than boys. Aside from this, we observed that most scores point to a different trend: boys were more in favor, or perceived their parents to be more in favor, of confidentiality and consent.

## A closer look at late adolescents' preferences

In Table 5, we present the results of four binomial regression models with respondent confidentiality and consent preferences as outcome variables and assumed parental attitudes and

**Table 3. Mean scores, paired samples *t*-test scores and *p*-values for each vignette.**

| | | Adolescent score | Parental score | *t*-test statistic | *p*-value |
|---|---|---|---|---|---|
| Confidentiality | Drunk (case 1) | .37 | .12 | 10.52 | .00 |
| | Std (case 3) | .68 | .28 | 16.42 | .00 |
| Consent | Surgery (case 2) | .79 | .63 | 7.02 | .00 |
| | Pill (case 4) | .77 | .45 | 13.44 | .00 |

Note. In confidentiality cases (drunk, std), a yes-answer indicates that the respondent is or believes their parents are of the opinion that the physician should report to the parents (regardless of legal regulations). In consent cases (surgery, pill), a yes-answer indicates that the respondent is or believes their parents are of the opinion that the physician should prescribe the pill/carry out the surgery, regardless of parental consent.

**Table 4. Mean scores, independent samples *t*-test scores and *p*-values for each vignette by gender.**

|  |  | **Boys** | **Girls** | ***t*-test statistic** | ***p*-value** |
|---|---|---|---|---|---|
| Confidentiality | Drunk$_a$ | .43 | .36 | 1.14 | .25 |
| | Drunk$_p$ | .14 | .11 | .66 | .51 |
| | Std$_a$ | .64 | .68 | -.67 | .51 |
| | Std$_p$ | .30 | .27 | .39 | .70 |
| Consent | Surgery$_a$ | .67 | .81 | -2.69 | .01 |
| | Surgery$_p$ | .61 | .64 | -.43 | .67 |
| | Pill$_a$ | .81 | .77 | .85 | .39 |
| | Pill$_p$ | .54 | .44 | 1.66 | .09 |

Note. High mean scores indicate high favorability of confidentiality (drunk, std) or non-parental consent (surgery, pill). The subscript a refers to respondents' own preferences. The subscript p refers to assumed parental preferences. Drunk = case 1, Surgery = case 2, Std = case 3, Pill = case 4.

controls as predictors. The results show that the association between respondents' own attitudes and their assumed parental attitudes was considerable, for all cases: we observed that the likelihood that respondents were in favour of consent/confidentiality was considerably higher when they believed their parents were also in favour of consent/confidentiality, as opposed to when they believe their parents were not in favour of consent/confidentiality. While controlling for age, gender, mothers' educational attainment, secondary educational level, and family situation, we found that respondents who believed that their parents supported their confidentiality or treatment without parental consent, would also be more in favor of these issues themselves. This was consistent for all cases. We also found that boys were significantly less likely than girls to want the maxillofacial surgery without their parents' consent, consistent with the results in Table 4.

## Discussion

As far as we know, this is the first study to investigate the extent to which attitudes of late adolescents concerning health care confidentiality and consent are related to their assumption of their own parents' views on these issues. We find that there is a clear discrepancy in preferences of late adolescents and what they believe their parents prefer. Adolescents value confidentiality and consent but assume that their parents differ in opinion. A possible difference between these assumptions may act as a barrier for adolescents to seek health care or decide on their own competence [6]. For example, late adolescents who contract with a sexually transmitted disease may delay or forgo health care if they believe their parents would disapprove of sexual behavior–regardless of their parents' actual opinion on this. Studies have shown that the actual difference in attitudes is less pronounced: parents hold similar attitudes to adolescents on issues of confidentiality [36], with adolescents slightly more in favor of confidentiality than parents. However, some studies have also shown that parents themselves sometimes hold ambivalent views regarding adolescents´ confidentiality [11–13]. They feel that confidential consultations endanger their ability to be a 'good parent' since they are partially excluded from their child's health care [11, 12]. None the less, our results show that parents' attitudes are important drivers of late adolescents' own attitudes: when late adolescents believe their parents are in favour of confidentiality/consent, they will also be more in favour of this themselves. When parents are then perceived to be against confidentiality, it is more likely that late adolescents will echo these sentiments with potential negative effects on their health-seeking behaviour as a consequence [6].

**Table 5. Binomial logistic regression with late adolescents' confidentiality and parental consent preferences as outcome variables, and assumed parental attitudes and controls as predictors.**

|  | Confidentiality | | Consent | |
|---|---|---|---|---|
|  | Drunk [a] (Case 1) | Std [a] (Case 3) | Surgery [b] (Case 2) | Pill [b] (Case 4) |
| **Assumed parental preference (ref: No consent/confidentiality preference)** |  |  |  |  |
| Consent/confidentiality preference | 1.66*** (0.35) | 2.84*** (0.47) | 2.57*** (0.30) | 2.47*** (0.37) |
| **Age** | 0.07 (0.12) | -0.11 (0.13) | 0.11 (0.14) | 0.09 (0.14) |
| **Gender (ref: girls)** |  |  |  |  |
| Boys | 0.22 (0.29) | -0.17 (0.32) | -0.84* (0.36) | 0.16 (0.39) |
| **Mothers' education level (ref: primary or secondary education)** |  |  |  |  |
| Tertiary education | -0.37 (0.25) | 0.09 (0.27) | 0.10 (0.33) | 0.18 (0.31) |
| **Educational level (ref: general education)** |  |  |  |  |
| Technical/vocational/artistic education | -0.14 (0.39) | 0.22 (0.45) | -0.19 (0.49) | -0.42 (0.45) |
| **Family situation (ref: intact family)** |  |  |  |  |
| Non-intact family | 0.05 (0.23) | 0.06 (0.26) | -0.40 (0.30) | -0.12 (0.28) |
| **Constant** | -1.72 (2.09) | 2.26 (2.30) | -1.77 (2.53) | -1.23 (2.62) |
| **Nagelkerke R$^2$** | .09 | .23 | .33 | .25 |

Note. *: $p < 0.05$;

**: $p < 0.01$,

***: $p < 0.001$.

[a] Reference category = 0 –Yes, inform parents regardless of adolescents' preferences.

[b] Reference category = 0 –No, do not proceed with treatment without parental consent.

The findings also point to another difference. Attitudes towards confidentiality and consent are also dependent on the type of (medical) situation. For cases on sexually transmitted infections and contraceptive use, the difference is more pronounced as respondents prefer physicians to respect their preferences concerning confidentiality or consent. The cases on drunkenness and jaw surgery were perceived with a somewhat larger degree of uniformity between late adolescents and parents. It is likely that the sensitive nature of the former two cases is the driving force behind the larger perceived attitude differences. Medical issues related to sexuality are sensitive topics among late adolescents and within families, which hinders an open discussion about sexual and reproductive health care [18, 19]. This ties into another possible explanation: a lack of knowledge about sexual and reproductive health care may reinforce respondents' preferences for confidentiality in those cases. Additionally, it is also possible that late adolescents who have a chronic condition or who have had health-care related experiences similar to the cases presented may answer differently than students who have had no similar experiences.

Gender differences show that boys are more in favor, or assume their parents to be more in favor, of confidentiality and consent. Previous studies have found that boys are more in favor of freedom in many areas of life and desire less parental presence during health assessments [18], while girls often place a higher value on the relationship with their parents [40]. Kappahan et al. (1999) investigated gender differences in confidentiality preferences and found that girls prefer to have a parent present during examinations, while boys had no preference [26]. As an exception to this trend, we found that boys are significantly less in favor of maxillofacial surgery without parental consent than girls. Further research must investigate these gender differences more thoroughly.

Our findings have potential limitations that we would like to discuss. Although they are novel, the sample is not representative of late adolescents in Flanders, Belgium. Since this is a convenience sample of first-year university students, it is important to consider this study a first foray into this field, and we invite other scholars and health care professionals to build on our insights

by establishing large-scale surveys among representative adolescent and adult samples to gain further insights into these attitude differences. And although the context of this study is local, it is also important to mention that several of our results align with findings of confidentiality studies in the United States, Canada and other parts of Europe that found that adolescents are–more than their parents–in favour of confidentiality and/or consent [21, 22, 27, 37], which indicates a universal aspect at the heart of these preferences irrespective of the legal framework in the individual country or state. None the less, follow-up research on this subject is paramount to further improve adolescent health care. The binomial logistic regressions revealed few differences by sociodemographic characteristics. This may not be due to the absence of an effect of such characteristics, but rather due to the homogeneous nature of our sample. Finally, since we used self-report measures, some of our data may be subject to social desirability bias (i.e. the tendency to answer questions in a way that will be viewed as favorable by others). Such bias is particularly known to occur when sensitive items have to be answered [41], which could be the case in this study with regards to the vignettes about sexually transmitted infections and contraceptive use.

We have opened several avenues of research into late adolescents' attitudes towards confidentiality and consent in health care. More research is needed on how the difference between late adolescents' own and assumed parental attitudes is associated with delaying or forgoing health-seeking behavior, as the potential ramifications of this for adolescent health may be considerable. Also, the lack of consistency between cases is interesting, as this suggests that individual preferences depend on the type of medical issue or the context. This is particularly important for issues related to sexual and reproductive health, which continue to be difficult to discuss in a family context. Health-care professionals can use this information to better inform parents and late adolescents. From a scholarly perspective, these context-dependent preferences indicate that confidentiality and consent are multifaceted concepts, and future studies should consider this complex structure when investigating them.

## Conclusion

Late adolescents hold different preferences towards confidentiality and consent than what they believe their parents prefer: in their opinion, health care professionals should respect late adolescents' confidentiality, while their parents are believed to be less inclined to feel this way. Late adolescents also prefer physicians to proceed with the treatment they prefer, regardless of parental consent. Understanding the difference between late adolescents' own and assumed parental preferences may provide further insight into late adolescents' health-seeking behavior.

## Supporting information

**S1 Appendix. English translation of vignettes.**
(DOCX)

**S2 Appendix. Full survey presented to late adolescents in Dutch.**
(DOCX)

**S1 Table. Descriptive overview of vignette answers (in %).**
(DOCX)

## Acknowledgments

We would like to take this opportunity to thank all study participants. We would also like to thank both anonymous reviewers for their highly valuable feedback on earlier versions of this article.

## Author Contributions

**Conceptualization:** David De Coninck, Koen Matthijs, Peter de Winter, Jaan Toelen.

**Formal analysis:** David De Coninck.

**Methodology:** David De Coninck, Koen Matthijs.

**Project administration:** Koen Matthijs, Peter de Winter.

**Supervision:** Jaan Toelen.

**Writing – original draft:** David De Coninck, Koen Matthijs, Peter de Winter, Jaan Toelen.

**Writing – review & editing:** Peter de Winter, Jaan Toelen.

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
