## [Decision Letter · Decision Letter 0]

24 Feb 2021

PONE-D-20-41061

Late Adolescents’ Own and Assumed Parental Preferences

Towards Health-Care Related Confidentiality and Competence in Belgium

PLOS ONE

Dear Dr. De Coninck,

Thank you for submitting your manuscript to PLOS ONE. After careful consideration, we feel that it has merit but does not fully meet PLOS ONE’s publication criteria as it currently stands. Therefore, we invite you to submit a revised version of the manuscript that addresses the points raised during the review process.

We look forward to receiving your revised manuscript.

Kind regards,

Ritesh G. Menezes, M.B.B.S., M.D., Diplomate N.B.

Academic Editor

PLOS ONE

Journal Requirements:

2. Please include additional information regarding the survey or questionnaire used in the study and ensure that you have provided sufficient details that others could replicate the analyses. For instance, if you developed a questionnaire as part of this study and it is not under a copyright more restrictive than CC-BY, please include a copy, in both the original language as well as the English version already provided, as Supporting Information.

4.You indicated that you had ethical approval for your study. In your Methods section, please ensure you have also stated whether you obtained consent from parents or guardians of the minors included in the study or whether the research ethics committee or IRB specifically waived the need for their consent.

5. Please list the name and version of any software package used for statistical analysis, alongside any relevant references. For more information on PLOS ONE's expectations for statistical reporting, please see https://journals.plos.org/plosone/s/submission-guidelines.#loc-statistical-reporting.

Reviewers' comments:

Reviewer's Responses to Questions

**Comments to the Author**

1. Is the manuscript technically sound, and do the data support the conclusions?

Reviewer #1: Partly

Reviewer #2: Yes

2. Has the statistical analysis been performed appropriately and rigorously? 

Reviewer #1: Yes

Reviewer #2: Yes

3. Have the authors made all data underlying the findings in their manuscript fully available?

Reviewer #1: No

Reviewer #2: No

4. Is the manuscript presented in an intelligible fashion and written in standard English?

Reviewer #1: Yes

Reviewer #2: No

5. Review Comments to the Author

Reviewer #1: RE: Late Adolescents’ Own and Assumed Parental Preferences Towards Health-Care Related Confidentiality and Competence in Belgium

The idea of this paper is very interesting, and novel. It seems important to compare adolescents’ perspective on confidentiality and consent with their perception of their own parent’s perspective on these issues. Discrepancy in these perspectives may point to the gap in adolescent-parent communication and problems that may arise in dealing with sensitive health issues. However, this paper needs major revisions in order to be suitable for publishing.

Introduction

1. The Introduction needs to be updated with more recent studies on consent, decision-making, and confidentiality in adolescent health care (most of the cited studies were published in 1990ies and early 2000s).

2. The importance of the role of the parents needs to be elaborated, since their perspective on confidentiality and consent is one of the research questions in this paper.

3. The authors need to clearly define main terms of the study: confidentiality and competence, in accordance with the well-recognized literature on medical ethics. “Competence” needs to be differentiated from the term “decision-making capacity”. Authors use the term “competence” for various concepts throughout the paper, although this term has clear definition and scope in the medical/legal/ethical field. I suggest changing it to “autonomy” or “consent”, since they better fit the actual concept that is inquired.

4. Aims of the study have to be clearly defined and listed at the and of the Introduction section. Please avoid sentences like “this study also aims to develop this field in Europe, more specifically in Belgium”, since this is too ambitious and goes far beyond your study.

5. Line 63: It is recommended to use references to original studies that you refer to.

6. Lines 79-80: Authors mention “several ethical, and personal considerations”, without explaining which are those considerations. If they are not to be listed, the sentence should be left out of the paper. Please avoid terms like “unfortunately”, since it is too colloquial.

7. Line 83 and 89: What does term “medically emancipated conditions” mean? Do you refer to the sensitive health issues? Please use terms that are widely accepted in the relevant literature in the field. In general, sexual and reproductive health, and mental health as well are considered as sensitive areas of health. The reason for granting minors right to confidential care in sensitive areas of health is to reduce harm (public health reason).

8. Line 103: Similar to previous: What are “competent decision”?

Materials and methods

1. All results and tables should be placed in the Results section. You should only describe methods of data collection, sampling, measures used and statistical analyses applied in the Material and Methods. When describing your sample, please include the description of the health care services that provide care to students. In some countries students are assigned to specialized health care facilities (students clinics) where the presence of parents is not required regardless of age. This would provide clearer picture of your respondents.

2. The main concepts you are investigating by vignettes are “confidentiality” and “competence”. In line with my previous comment, please consider re-naming competence, since it is a strictly defined term referring to legal standard. Since your vignettes inquire respondents’ belief on whether the physician should prescribe the pill over parental objection and belief on whether adolescent should be allowed to make independent decision on the surgery, I suggest you to label this concept “autonomy” or “consent”. In fact your questions do not inquire the issue of decision-making capacity and competence.

3. Since you asked only for mother’s educational status, it cannot be used as a proxy for SES status. SES status is calculated based on more variables (such as income, housing conditions, education, employment etc.).

Results

1. In the supplement material you provided original questions in the vignettes. In the vignette 1 the question for the respondents is: Do you believe that the physician has the right to report the ‘drunkenness’ to your parents, despite your express request not to do so?

A question worded like this asks from respondents to express their belief regarding the physician’s right to report to parents despite the adolescent’s quest for confidentiality. It does not reveal respondent’s preference or choice regarding confidentiality, so it shouldn’t be presented and discussed as respondent’s preference toward confidentiality. Similar is the situation with the vignette 3. This is the largest objection to this paper.

In the description of measures (Lines 135-136) you stated: “To gauge respondents’ own and assumed parental preferences towards confidentiality and competence, we present them with four fictional cases”, (lines 138-139): “For each case, the respondent was asked to indicate their own preference regarding confidentiality or competence.”

The fact is, according to original vignettes, respondents were indicating their belief on whether the physician has the right to report drunkenness to parents or the right to inform the parents about the STD. Since original vignettes were in Dutch, maybe the translation should be checked. Anyway, the text in the vignettes should correspond to the text in the Methods and Results.

Having in mind original text in vignettes, you cannot report your results as such: (lines 199-204): “The results in Table 3 indicate that respondents have significantly different preferences towards confidentiality and competence than what they believe their parents have. The mean scores show that with regard to the cases on drunkenness and the sexually transmitted disease, which both frame issues of confidentiality, respondents expect that their parents want to be informed (drunk: μ = .12, SD = .32 / std: μ = .28, SD = .45), while respondents themselves would prefer the health care professional to respect the confidentiality”.

What you CAN say for example is that respondents believe their parents want to be informed, while they believe physician doesn’t have the right to report to parents.

In accordance with this suggestion, you also need to reformulate presentation of the regression results.

Discussion

1. You base your discussion on the misinterpretation of results. You need to be more precise and stick to your data when drawing conclusions. You cannot conclude about respondent’s preferences toward confidentiality from your vignettes, since they are not investigating preferences toward confidentiality but respondents’ beliefs regarding physician’s rights to tell the parents. It may happen that the respondent believes that the physician has the right to tell the parents, and at the same time feels that this is not fair, and that adolescent should have confidentiality guaranteed. You need to correct such interpretation throughout the whole paper, and especially in making conclusions. Also, the Title should be changed, to be in line with actual results of the study.

2. Lines 279-282: “And although the context of this study is local, it is also important to mention that several of our results align with findings of confidentiality studies in the United States, Canada and other parts of Europe [6,7,15,17], which indicates a universal aspect at the heart of these preferences irrespective of the legal framework in the individual country or state.”

It is not obvious to readers how your results align with the findings of other studies. You need to relate your results to the results of similar studies in more details.

3. Lines 286-288: “Finally, since we used self-report measures, some of our data may be subject to reporting bias (e.g., see the discussion on the sensitive nature of some cases and individual experiences)”

Which discussion? You need to explain in brief, and provide the reference.

4. Lines 289-290: “The casuistic methodology we applied is not often employed in studies on confidentiality. In most cases, respondents are asked to evaluate the (abstract) concepts of confidentiality or competence on Likert scales, resulting in subjective evaluations which threaten the external validity of these concepts. Making such abstract concepts more concrete for respondents through specific cases ensures a more reliable measurement of confidentiality and competence preferences in real life scenarios “

Actually, there are many studies that use cases and vignettes, and also qualitative studies in the field of adolescent confidentiality. Moreover, Likert-based instruments can also be built to have satisfying external validity. Also, you didn’t apply any assessment of validity and reliability in your instrument, so I suggest to leave this paragraph from the paper.

Reviewer #2: Thank you for the opportunity to review this paper regarding adolescents´ own and assumed parental preferences regarding confidentiality and competence in Belgium. The study deals with a clinically important and difficult issue when treating adolescents and the authors have chosen a novel and innovative approach. The authors find a discrepancy between the adolescents´ own and assumed parental preferences in all four vignettes as well as a strong association between their own and their assumed parental preference which may affect adolecents´ health seeking behavior and thus have a major clinical impact.

Major/general comments:

Although the manuscript is well-written, the authors switch between present and past tense throughout the manuscript and there are some grammatical errors and typos. Furthermore, there are some words that may not be the most descriptive and thus could be changed i.e. disjunction. Thus, I recommend a linguistic review by a English native speaking person.

Many of the references used are of older age and may be replaced by newer alternatives. E.g. ref 26 regarding brain development is from 2009! And more than half of references are from before 2010.

The background section may be shortened and worded more stringent

The Material and Methods section seems a bit unorganized and could be more stringent

The Results section would profit from starting with general text and not just a table.

I think that the differences in the two vignettes concerning consent should be discussed more unless there is a mistake in the table 1A in the supplementary material

Limitations section:

Some of the vignettes and especially the ‘surgery’ seems quite specific and may need some background knowledge from the participants.

I am not sure I understand the issue of ‘reporting bias’ (line 287) as the participants assessed vignettes and not reporting their own experiences. And I do not understand the text in parentheses (line 287) regarding the discussion on the sensitive nature of some cases.

Whether the participants suffer from a chronic condition and/or have their own experience with health care professionals may impact their replies and this could be discussed in the limitations section

Minor comments

Please be consistent in using ‘vignette’ or ‘case’ as well as the names of the vignettes (e.g. surgery vs. maxi)

Line 55-56: rather than using this quotation, I would use the argument of adolescent brain and psychosocial development as well as different legislation in different countries to introduce the issue.

Line 110: rather than the parents´ actual perceptions the study the participants were asked of their´ assumed parental perceptions

Line 163: gender: was it possible to tick other options than male/female?

Line 167: cohabitation options: where there other options e.g. widow?

line 176-179: the argument of brain development is better placed somewhere else than in the Material and Methods section.

Line 184-193 should be placed in a ‘statistics section’ and not starting with ‘subsequently…’

Table 2 is described in the Methods section but the actual table is placed under Results which is quite confusing. Furthermore, the table needs a legend with some explanation

Line 199-213: the text seems more complicated than needed and basically report the results shown in table 3. Thus, this paragraph could be shortened.

Table 3: I notice a very high SD that the authors may discuss (e.g. drunk (a) .37 (.48), drunk (p) .12 (.32))

The vignette ‘surgery’ changed its name to ‘maxi’

Table 4: I notice a very high SD that the authors may discuss

Line 227-233: The authors may discuss the issue of considerable difference between participants preferences and their assumed parental attitudes and at the same time a considerable association between participants´ own attitudes and their assumed parental attitudes

Line 233-235 the sentence is better placed together with the paragraph on gender issues (line 218-222)

Line 238: replace ‘young adults´’ with ‘adolescents’ as you argue previously

Table 5: I struggle a bit understanding this table and the lines seems a bit messy. The authors introduce a new concept (autonomy with adolescent?)

Discussion:

Line 249-253: You may add that parents themselves may hold ambivalent views regarding adolescents´ confidentiality (e.g. Duncan RE et al. JAH 2011, Sasse RA et al, JAH 2013, Thomsen EL et al. IJAMH 2019) to the discussion

Line 265-267: please give a reference to this statement (boys are more in favor of freedom)

Line 267: Kappahan is a quite aged reference and this gender difference may have changed

Line 270-271: the preunderstanding of the meaning of ‘looks’ and gender seems somewhat dated

6. PLOS authors have the option to publish the peer review history of their article (what does this mean?). If published, this will include your full peer review and any attached files.

Reviewer #1: No

Reviewer #2: No

---

## [Author Response · Author response to Decision Letter 0]

10 Mar 2021

Dear Editor,

Dear Prof. Menezes,

Thank you for your feedback on our paper, which was very helpful. In this letter, we provide a point-by-point response to each of the comments and suggestions you and the reviewers provided. We hope that this version can be reconsidered for publication in PLOS One.

With warm regards,

David

Editor/journal comments

Comment 1: Please ensure that your manuscript meets PLOS ONE's style requirements, including those for file naming. The PLOS ONE style templates can be found at

Response: We have revised our files in line with these requirements. 

Comment 2: Please include additional information regarding the survey or questionnaire used in the study and ensure that you have provided sufficient details that others could replicate the analyses. For instance, if you developed a questionnaire as part of this study and it is not under a copyright more restrictive than CC-BY, please include a copy, in both the original language as well as the English version already provided, as Supporting Information.

Response: We have now included a version of the original Dutch survey used in the supplemental materials, in addition to the information already provided regarding the English translation.

Comment 3: We note that you have indicated that data from this study are available upon request. PLOS only allows data to be available upon request if there are legal or ethical restrictions on sharing data publicly. For information on unacceptable data access restrictions, please see http://journals.plos.org/plosone/s/data-availability#loc-unacceptable-data-access-restrictions.

Response: As outlined in the ethical approval of the Social and Societal Ethics Committee of the KU Leuven (case number G-2017 09 934), we are not allowed to publicly share the dataset given that it contains data on KU Leuven students. Data requests may be sent to the authors, who may share the data on a case-by-case basis. Please contact the corresponding author for these requests (David.deconinck@kuleuven.be).

Comment 4: You indicated that you had ethical approval for your study. In your Methods section, please ensure you have also stated whether you obtained consent from parents or guardians of the minors included in the study or whether the research ethics committee or IRB specifically waived the need for their consent.

Response: Given that all underage participants were 17 years old and active students at the KU Leuven, the ethical committee did not require us to obtain informed consent from parents or guardians. 

Comment 5. Please list the name and version of any software package used for statistical analysis, alongside any relevant references. For more information on PLOS ONE's expectations for statistical reporting, please see https://journals.plos.org/plosone/s/submission-guidelines.#loc-statistical-reporting.

Response: We now report the software package and version used for all analyses. 

Comment 6: Your ethics statement should only appear in the Methods section of your manuscript. If your ethics statement is written in any section besides the Methods, please delete it from any other section.

Response: This has been done.

Comment 7: Please include captions for your Supporting Information files at the end of your manuscript, and update any in-text citations to match accordingly. Please see our Supporting Information guidelines for more information: http://journals.plos.org/plosone/s/supporting-information.

Response: We have updated our supporting information captions at the end of our manuscript and have matched the in-text citations accordingly in line with the Plos One-guidelines.

 

Reviewer 1

Comment 1: The idea of this paper is very interesting, and novel. It seems important to compare adolescents’ perspective on confidentiality and consent with their perception of their own parent’s perspective on these issues. Discrepancy in these perspectives may point to the gap in adolescent-parent communication and problems that may arise in dealing with sensitive health issues. However, this paper needs major revisions in order to be suitable for publishing.

Response: Thank you for your positive and constructive feedback on our manuscript. Below, you can find a point-by-point response to your comments, which helped us create a better article. We hope this can be again considered for publication in Plos One. 

Comment 2: The Introduction needs to be updated with more recent studies on consent, decision-making, and confidentiality in adolescent health care (most of the cited studies were published in the 1990s and early 2000s). 

Response: While this specific subject is a difficult one to find more recent studies about – many studies regarding confidentiality focus on physicians’ perspectives – we have found some additional empirical material from and more recently 2010 (also with thanks to Reviewer 2’s comments) and have incorporated these references into our introduction and discussion. 

Comment 3: The importance of the role of the parents needs to be elaborated since their perspective on confidentiality and consent is one of the research questions in this paper. 

Response: We agree that this perspective was a bit understated in the previous version of the introduction. We have now added a brief overview of the parental perspective (see line 75-82), along with some recent references. 

Comment 4: The authors need to clearly define main terms of the study: confidentiality and competence, in accordance with the well-recognized literature on medical ethics. “Competence” needs to be differentiated from the term “decision-making capacity”. Authors use the term “competence” for various concepts throughout the paper, although this term has clear definition and scope in the medical/legal/ethical field. I suggest changing it to “autonomy” or “consent”, since they better fit the actual concept that is inquired.

Response: We thank the reviewer for this valuable comment. We actually struggled with this same issue in writing the article and had used the term “consent” instead of “competence” in earlier versions of this manuscript. With your feedback in mind, I believe we are indeed much better off using the term “consent” once again. This has been amended throughout.

Comment 5: Aims of the study have to be clearly defined and listed at the and of the Introduction section. Please avoid sentences like “this study also aims to develop this field in Europe, more specifically in Belgium”, since this is too ambitious and goes far beyond your study.

Response: We have removed the section you were referring to as being too ambitious, as this is indeed beyond the scope of our study. We have also tried to clearly articulate and presents the aims of our study in that same paragraph. 

Comment 6: Line 63: It is recommended to use references to original studies that you refer to.

Response: These references have been added. 

Comment 7: Lines 79-80: Authors mention “several ethical, and personal considerations”, without explaining which are those considerations. If they are not to be listed, the sentence should be left out of the paper. Please avoid terms like “unfortunately”, since it is too colloquial.

Response: We agree with this. We removed this sentence and also took out the word ‘unfortunately’. 

Comment 8: Line 83 and 89: What does term “medically emancipated conditions” mean? Do you refer to the sensitive health issues? Please use terms that are widely accepted in the relevant literature in the field. In general, sexual and reproductive health, and mental health as well are considered as sensitive areas of health. The reason for granting minors right to confidential care in sensitive areas of health is to reduce harm (public health reason). 

Response: We understand the confusion regarding the terminology ‘medically emancipated conditions’. Because this line originates from a quote, we decided not to remove it, but we have added some extra context between brackets so that readers better understand what is understood by such conditions. More specifically, we added: [i.e. sensitive health issues, like sexual or reproductive health]. In line 89, we have removed the reference to medically emancipated conditions.

Comment 9: Line 103: Similar to previous: What are “competent decision”?

Response: Given that we now talk about consent instead of competence throughout the article, we needed to rewrite this sentence somewhat. Instead, we now say: “While health-care related decisions should be free of biases like false assumptions, misinformation […]”. 

Materials and methods 

Comment 10: All results and tables should be placed in the Results section. You should only describe methods of data collection, sampling, measures used and statistical analyses applied in the Material and Methods. When describing your sample, please include the description of the health care services that provide care to students. In some countries students are assigned to specialized health care facilities (students clinics) where the presence of parents is not required regardless of age. This would provide clearer picture of your respondents.

Response: We have now moved all tables to the Results-section and restructured this section somewhat. We now use 3 subsections (descriptive overview, t-test results, regression results) to better structure the results. In Belgium, students have no special health care services that provide care to students, aside from university-wide initiatives that provide mental health support. Because nearly all native students commute home during weekends, they often retain the same health care services as they did prior to their university enrolment. 

Comment 11: The main concepts you are investigating by vignettes are “confidentiality” and “competence”. In line with my previous comment, please consider re-naming competence, since it is a strictly defined term referring to legal standard. Since your vignettes inquire respondents’ belief on whether the physician should prescribe the pill over parental objection and belief on whether adolescent should be allowed to make independent decision on the surgery, I suggest you tolabel this concept “autonomy” or “consent”. In fact your questions do not inquire the issue of decision-making capacity and competence. 

Response: In line with your previous comment, we have now changed all instances of “competence” to “consent”. 

Comment 12: Since you asked only for mother’s educational status, it cannot be used as a proxy for SES status. SES status is calculated based on more variables (such as income, housing conditions, education, employment etc.).

Response: We agree with this comment and have removed the reference to the SES in this sentence.

Results

Comment 13: In the supplement material you provided original questions in the vignettes. In the vignette 1 the question for the respondents is: Do you believe that the physician has the right to report the ‘drunkenness’ to your parents, despite your express request not to do so?

A question worded like this asks from respondents to express their belief regarding the physician’s right to report to parents despite the adolescent’s quest for confidentiality. It does not reveal respondent’s preference or choice regarding confidentiality, so it shouldn’t be presented and discussed as respondent’s preference toward confidentiality. Similar is the situation with the vignette 3. This is the largest objection to this paper. 

In the description of measures (Lines 135-136) you stated: “To gauge respondents’ own and assumed parental preferences towards confidentiality and competence, we present them with four fictional cases”, (lines 138-139): “For each case, the respondent was asked to indicate their own preference regarding confidentiality or competence.” 

The fact is, according to original vignettes, respondents were indicating their belief on whether the physician has the right to report drunkenness to parents or the right to inform the parents about the STD. Since original vignettes were in Dutch, maybe the translation should be checked. Anyway, the text in the vignettes should correspond to the text in the Methods and Results.

Having in mind original text in vignettes, you cannot report your results as such: (lines 199-204): “The results in Table 3 indicate that respondents have significantly different preferences towards confidentiality and competence than what they believe their parents have. The mean scores show that with regard to the cases on drunkenness and the sexually transmitted disease, which both frame issues of confidentiality, respondents expect that their parents want to be informed (drunk: μ = .12, SD = .32 / std: μ = .28, SD = .45), while respondents themselves would prefer the health care professional to respect the confidentiality”. 

What you CAN say for example is that respondents believe their parents want to be informed, while they believe physician doesn’t have the right to report to parents.

In accordance with this suggestion, you also need to reformulate presentation of the regression results.

Response: We thank you for your very important comment regarding the vignettes. Fundamentally, I believe this is indeed an issue of translation rather than a conceptual flaw within the vignettes. We very much gauged the adolescents’ opinion about what the physician should do, rather than what they believe he/she is legally obligated or allowed to do. Thus, your comments highlighted a flaw in our translation and conceptual clarity, and we thank you for this. We have revised the translation of the vignettes in the supplemental materials and checked these with a native English speaker. We have also revised the text in the Methods regarding the vignettes to align with the new translation. 

Discussion

Comment 14: You base your discussion on the misinterpretation of results. You need to be more precise and stick to your data when drawing conclusions. You cannot conclude about respondent’s preferences toward confidentiality from your vignettes, since they are not investigating preferences toward confidentiality but respondents’ beliefs regarding physician’s rights to tell the parents. It may happen that the respondent believes that the physician has the right to tell the parents, and at the same time feels that this is not fair, and that adolescent should have confidentiality guaranteed. You need to correct such interpretation throughout the whole paper, and especially in making conclusions. Also, the Title should be changed, to be in line with actual results of the study.

Response: We have changed the title to reflect the consent-terminology used throughout the paper. Your comments about the discussion are, of course, valid but our revised translation of the vignettes should counter some of these perceived conceptual misinterpretations.

Comment 15: Lines 279-282: “And although the context of this study is local, it is also important to mention that several of our results align with findings of confidentiality studies in the United States, Canada and other parts of Europe [6,7,15,17], which indicates a universal aspect at the heart of these preferences irrespective of the legal framework in the individual country or state.”

It is not obvious to readers how your results align with the findings of other studies. You need to relate your results to the results of similar studies in more details.

Response: We agree with your comment, and we have expanded this sentence to align which results of these studies align with ours.

Comment 16: Lines 286-288: “Finally, since we used self-report measures, some of our data may be subject to reporting bias (e.g., see the discussion on the sensitive nature of some cases and individual experiences)” Which discussion? You need to explain in brief, and provide the reference.

Response: We agree that we did not clarify this enough in this sentence. We’ve expanded this section and hope our arguments are now clearer.

Comment 17: Lines 289-290: “The casuistic methodology we applied is not often employed in studies on confidentiality. In most cases, respondents are asked to evaluate the (abstract) concepts of confidentiality or competence on Likert scales, resulting in subjective evaluations which threaten the external validity of these concepts. Making such abstract concepts more concrete for respondents through specific cases ensures a more reliable measurement of confidentiality and competence preferences in real life scenarios“ 

Actually, there are many studies that use cases and vignettes, and also qualitative studies in the field of adolescent confidentiality. Moreover, Likert-based instruments can also be built to have satisfying external validity. Also, you didn’t apply any assessment of validity and reliability in your instrument, so I suggest to leave this paragraph from the paper.

Response: We agree with this comment and have removed the paragraph from the paper. 

Reviewer 2

Comment 1: Thank you for the opportunity to review this paper regarding adolescents´ own and assumed parental preferences regarding confidentiality and competence in Belgium. The study deals with a clinically important and difficult issue when treating adolescents and the authors have chosen a novel and innovative approach. The authors find a discrepancy between the adolescents´ own and assumed parental preferences in all four vignettes as well as a strong association between their own and their assumed parental preference which may affect adolecents´ health seeking behavior and thus have a major clinical impact.

Response: Thank you for your positive and constructive feedback on our manuscript. Below, you can find a point-by-point response to your comments, which helped us create a better article. We hope this can be again considered for publication in Plos One. 

Major comments

Comment 2: Although the manuscript is well-written, the authors switch between present and past tense throughout the manuscript and there are some grammatical errors and typos. Furthermore, there are some words that may not be the most descriptive and thus could be changed i.e. disjunction. Thus, I recommend a linguistic review by a English native speaking person.

Response: We have thoroughly proofread the paper, and now use the present tense in both the introduction and discussion and use the past tense in the methodology and results, as is common in most academic papers. We’ve also removed the word ‘disjunction’, and either rewrote the sentence so we did not need to use it or used the word ‘difference’. Further language revisions were conducted following a check by a native English speaker.

Comment 3: Many of the references used are of older age and may be replaced by newer alternatives. E.g. ref 26 regarding brain development is from 2009! And more than half of references are from before 2010.

Response: While this specific subject is a difficult one to find more recent studies about – most studies regarding confidentiality focus on physicians’ perspectives – we have found additional empirical material from 2010 and more recently and have incorporated these references into our introduction. We have also added an extra reference to the point regarding brain development. 

Comment 4: The background section may be shortened and worded more stringent

The Material and Methods section seems a bit unorganized and could be more stringent

Response: We have thoroughly restructured the Methods and Results section. Information about the sample is now moved to the Results (descriptive analyses), while the methods are now strictly for discussing the methodology and data collection, and outlining the measures used. We have also cut down several sections in the introduction where possible/appropriate.

Comment 5: The Results section would profit from starting with general text and not just a table.

Response: Thank you for this suggestion, we have restructured the methods and results-section somewhat and now start with general text.

Comment 6: I think that the differences in the two vignettes concerning consent should be discussed more unless there is a mistake in the table 1A in the supplementary material

Response: While there is indeed a descriptive difference between those two vignettes, it is actually more of a coding issue than actual differences in attitudes between vignettes. In Vignette 3 (std), a yes-answer indicates that the participant is of the opinion that the physician should report the std to the parents, regardless of their consent. Thus, the yes-answer is ‘anti-consent’ in this case, sort of speak. In Vignette 4 (pill), a yes-answer indicates that the participant is of the opinion that the physician should prescribe the pill, regardless of consent. Thus, the yes-answer is ‘pro-consent’ in this case. So the fact that the descriptive results are mirrored between these two cases actually indicates a fair degree of consistency in answers. 

Limitations section: 

Comment 7: Some of the vignettes and especially the ‘surgery’ seems quite specific and may need some background knowledge from the participants.

Response: We have tried to formulate the vignettes in such a way that they were widely understandable. No specific medical jargon was used in any of them, allowing participants to understand the vignettes easily. We also pre-tested these vignettes among a limited sample of friends/family. While this is not representative in any way of course, we received no remarks on the wording of the vignettes.

Comment 8: I am not sure I understand the issue of ‘reporting bias’ (line 287) as the participants assessed vignettes and not reporting their own experiences. And I do not understand the text in parentheses (line 287) regarding the discussion on the sensitive nature of some cases.

Response: This is in line with comments from Reviewer 1. We have revised this section of the discussion, which should hopefully clear up some of the confusion. 

Comment 9: Whether the participants suffer from a chronic condition and/or have their own experience with health care professionals may impact their replies and this could be discussed in the limitations section

Response: This is an excellent suggestion and may very well be the case. We have incorporated this into the discussion. 

Minor comments 

Comment 10: Please be consistent in using ‘vignette’ or ‘case’ as well as the names of the vignettes (e.g. surgery vs. maxi) 

Response: We have rewritten some sentences to align with your request, but we also felt that changing the terminology somewhat increased readability for some sentences. However, when we first describe the vignettes (see line 149) we now state that we also use the term ‘cases’ to refer to these vignettes in the rest of the article, so that the reader clearly know that we mean the same thing when we talk about ‘cases’ and ‘vignettes’. 

Comment 11: Line 55-56: rather than using this quotation, I would use the argument of adolescent brain and psychosocial development as well as different legislation in different countries to introduce the issue.

Response: We agree that the quotation was not the best opening section and decided to remove it. We believe the introduction reads better without it. We opted against moving the discussion on psychosocial development to the beginning, but we have moved the section regarding the different legislations to the very top and coupled that with the new section on parental attitudes towards confidentiality. We believe this restructuring benefits the introduction, so thank you for this suggestion. 

Comment 12: Line 110: rather than the parents´ actual perceptions the study the participants were asked of their´ assumed parental perceptions 

Response: Thank you for this suggestion, we have incorporated it into our paragraph.

Comment 13: Line 163: gender: was it possible to tick other options than male/female?

Response: There were no additional options available. 

Comment 14: Line 167: cohabitation options: where there other options e.g. widow?

Response: While students were unable to indicate if their parent was a widow/widower, we did inquire whether (one of) their biological parents had passed away in a separate question. However, none of the students in our sample indicated this to be the case. 

Comment 15: line 176-179: the argument of brain development is better placed somewhere else than in the Material and Methods section. 

Response: We have moved this argument to the end of the introduction, when we specify the aims and main research question.

Comment 16: Line 184-193 should be placed in a ‘statistics section’ and not starting with ‘subsequently…’

Response: Following a restructuring of the Methods and Results section, this section is now located under the ‘Results’ heading. We’ve also rewritten the sentence to remove the word ‘subsequently’.

Comment 17: Table 2 is described in the Methods section but the actual table is placed under Results which is quite confusing. Furthermore, the table needs a legend with some explanation

Response: Following the restructuring of the Methods and Results, both the description and Table 2 are located in the Results-section. 

Comment 18: Line 199-213: the text seems more complicated than needed and basically report the results shown in table 3. Thus, this paragraph could be shortened.

Response: We have revised this paragraph and cut it down where possible.

Comment 19: Table 3: I notice a very high SD that the authors may discuss (e.g. drunk (a) .37 (.48), drunk (p) .12 (.32)). Table 4: I notice a very high SD that the authors may discuss

Response: We thank the reviewer for their keen eye with regards to the SDs. Because these cases are dummy variables (yes/no), the interpretation of the SD is somewhat different and less meaningful than it is for metric/numeric or ordinal variables. The SD is the difference between each data point from the mean. So, since the mean of a dichotomous variable is the per cent who were coded 1 (as a decimal), the standard deviation would be the difference from 1 to that decimal for everyone who responded 1 (squared and summed) combined with the difference from 0 to that decimal for everyone who responded 0 (squared and summer), then divided by the number of responses (minus 1) and square-rooted. While this information is interesting for metric variables, it is much less so for dummies. We therefore decided to remove the SDs from the tables, which also increases the readability of some of the results. 

Comment 20: The vignette ‘surgery’ changed its name to ‘maxi’

Response: We have corrected this. 

Comment 21: Line 227-233: The authors may discuss the issue of considerable difference between participants preferences and their assumed parental attitudes and at the same time a considerable association between participants´ own attitudes and their assumed parental attitudes

Response: We have incorporated this into the discussion.

Comment 22: Line 233-235 the sentence is better placed together with the paragraph on gender issues (line 218-222) 

Response: Following our restructuring of the Results, this sentence is part of the explanation of the logistic regression (under this subheading) – moving it to the t-test section of the results may confuse readers.

Comment 23: Line 238: replace ‘young adults´’ with ‘adolescents’ as you argue previously

Response: Thank you for noticing this oversight. This has been amended.

Comment 24: Table 5: I struggle a bit understanding this table and the lines seems a bit messy. The authors introduce a new concept (autonomy with adolescent?)

Response: Upon re-reading this section, we understand that the presentation may seem somewhat confusing. We have expanded the interpretation of the results to hopefully clarify them further. We also removed the ‘autonomy with adolescent’ terminology and replaced it with terminology that is used throughout the article.

Discussion:

Comment 25: Line 249-253: You may add that parents themselves may hold ambivalent views regarding adolescents´ confidentiality (e.g. Duncan RE et al. JAH 2011, Sasse RA et al, JAH 2013, Thomsen EL et al. IJAMH 2019) to the discussion 

Response: These are some excellent suggestions for additional literature, and we have incorporated their insights in the discussion and the introduction. Thank you for this!

Comment 26: Line 265-267: please give a reference to this statement (boys are more in favor of freedom)

Response: We have now added a recent source (from 2017) to support this statement.

Comment 27: Line 267: Kappahan is a quite aged reference and this gender difference may have changed

Response: We have added a more recent reference from 2014 to further support this statement.

Comment 28: Line 270-271: the preunderstanding of the meaning of ‘looks’ and gender seems somewhat dated

Response: We agree with this and have removed it from the discussion.

---

## [Decision Letter · Decision Letter 1]

13 Apr 2021

PONE-D-20-41061R1

Late Adolescents’ Own and Assumed Parental Preferences Towards Health-Care Related Confidentiality and Consent in Belgium

PLOS ONE

Dear Dr. De Coninck,

Thank you for submitting your manuscript to PLOS ONE. After careful consideration, we feel that it has merit but does not fully meet PLOS ONE’s publication criteria as it currently stands. Therefore, we invite you to submit a revised version of the manuscript that addresses the points raised during the review process.

We look forward to receiving your revised manuscript.

Kind regards,

Prof. Ritesh G. Menezes, M.B.B.S., M.D., Diplomate N.B.

Academic Editor

PLOS ONE

Journal Requirements:

Additional Academic Editor Comments:

Kindly note that some of the comments made by the reviewers on the previous version are not appropriately addressed in the revised version. Please do have a relook at the previous set of comments in addition to the new set of comments.

Reviewers' comments:

Reviewer's Responses to Questions

**Comments to the Author**

1. If the authors have adequately addressed your comments raised in a previous round of review and you feel that this manuscript is now acceptable for publication, you may indicate that here to bypass the “Comments to the Author” section, enter your conflict of interest statement in the “Confidential to Editor” section, and submit your "Accept" recommendation.

Reviewer #1: (No Response)

Reviewer #2: (No Response)

2. Is the manuscript technically sound, and do the data support the conclusions?

Reviewer #1: Yes

Reviewer #2: Yes

3. Has the statistical analysis been performed appropriately and rigorously? 

Reviewer #1: Yes

Reviewer #2: Yes

4. Have the authors made all data underlying the findings in their manuscript fully available?

Reviewer #1: No

Reviewer #2: Yes

5. Is the manuscript presented in an intelligible fashion and written in standard English?

Reviewer #1: Yes

Reviewer #2: Yes

6. Review Comments to the Author

Reviewer #1: Re: Late Adolescents’ Own and Assumed Parental Preferences Towards Health-Care

Related Confidentiality and Consent in Belgium

I want to congratulate the authors on the significant improvement of the paper. The authors addressed all comments from the review in scientifically sound and intelligible way. The paper is now suitable for publication. There are only few very small comments that need to be addressed before publishing.

Minor comments

1. Lines 191-192, 195-200: The description of analyses should be placed in the Methods section, under the Data analysis header. The results of analyses should remain in the Results section.

2. Line 271: a word is missing (contraceptive use, the is more pronounced as respondents prefer physicians to respect their preferences)

3. You don’t need to provide the full questionnaire in Dutch, it is sufficient to have the section that was described in the paper, in English.

Reviewer #2: Thank you for the opportunity to review this revised paper regarding adolescents´ own and assumed parental preferences regarding confidentiality and consent in Belgium. This is an important and well-performed study dealing with a clinically important and difficult issue when treating adolescents. The authors manuscript has improved significantly, and I have only minor comments.

In the abstract, methods and results are still written in present tense and not past tense as in the main manuscript.

Background:

Line 80-85: you state that evidence from Europe is lacking. However, Thomsen et al have looked at parents’ ambivalent feelings regarding confidentiality in a paper from 2019. The reference from Thomsen, that you do cite is dealing with guidelines for adolescent friendly health services and not only confidentiality and consent.

Line 96-98: I recommend you to use more recent guidelines regarding time alone during consultations

Line 105-109:Again I recommend newer references e.g.Mcdonagh et al, Acta Paediatrica 2006

Line 113+: the aim of the study should probably be in past tense

Line 123-131: I think that this highly relevant paragraph should be moved up in the background section

Materials and methods:

Line 139: I guess the students were asked to participate in the study and not in class?

Line 142-147: I think this should be moved to an Ethics section

I miss a statistics section and suggest that you re-phrase and move the paragraph line 191-200 from the results section

Results:

Line 205: please provide a more descriptive headline than T test results.

Line 209-10: You mention SD in the ‘drunk’ case

Line 206-218: This section is a repetition from the table 3 and I think you could omit the numbers and refer to the table. Furthermore, I think that your explanation in your reply to the reviewers regarding how you get the results should be included in the text or table legend as it makes it much more easy to understand.

Line 223: I think it is an overstatement to say that there is limited significant differences as it only applies to the ‘surgery case’ and this could be ‘by chance’. If you think the difference is true, I suggest you discuss this in the discussion section as it is quite interesting.

Line232: we present (present tense)

Line 236-237: believed (past tense)

Line 240 supported (past tense)

Discussion:

Line 253: it seems that there are words missing e.g. ‘perceptions between….’

Line 262: I would use the reference by Thomsen et al describing parents’ ambivalent views

Line 271: is there a word missing?

Conclusion

Line 322: regardless of parental consent?

7. PLOS authors have the option to publish the peer review history of their article (what does this mean?). If published, this will include your full peer review and any attached files.

Reviewer #1: No

Reviewer #2: No

---

## [Author Response · Author response to Decision Letter 1]

13 Apr 2021

Dear Editor,

Dear Prof. Menezes,

Thank you for your feedback on our paper, which was very helpful. We would also like to thank both reviewers for their positive feedback on the newest version of our manuscript. In this letter, we provide a point-by-point response to each of the remaining comments and suggestions you and the reviewers provided. We hope that this version can be reconsidered for publication in PLOS One.

With warm regards,

David

Editor/journal comments

Comment 1: Kindly note that some of the comments made by the reviewers on the previous version are not appropriately addressed in the revised version. Please do have a relook at the previous set of comments in addition to the new set of comments.

Response: Thank you for this comment. We have revisited the earlier comments from the reviewers and have made some additional revisions with regards to the tense of our manuscript. Aside from this, we believe we have thoroughly addressed all comments from the reviewers – but are willing to revisit specific points if need be.

 

Reviewer 1

Comment 1: I want to congratulate the authors on the significant improvement of the paper. The authors addressed all comments from the review in scientifically sound and intelligible way. The paper is now suitable for publication. There are only few very small comments that need to be addressed before publishing.

Response: Thank you for your positive feedback on the revised version of our manuscript. Below, you can find a point-by-point response to your minor comments. We hope this version can be accepted for publication in Plos One. 

Minor comments: 

Comment 2: Lines 191-192, 195-200: The description of analyses should be placed in the Methods section, under the Data analysis header. The results of analyses should remain in the Results section.

Response: In line with your comments, we have now created a new header ‘Data analysis’ (just before the Results section, still in the Methods section), which contains the information from the lines you highlighted. 

Comment 3: Line 271: a word is missing (contraceptive use, the is more pronounced as respondents prefer physicians to respect their preferences).

Response: Thank you for noticing this! You are quite right – the word ‘difference’ was missing. We have added it. 

Comment 4: You don’t need to provide the full questionnaire in Dutch, it is sufficient to have the section that was described in the paper, in English.

Response: We have removed the superfluous sections of the survey that were provided, and now only present the Dutch version of the vignettes in the supplemental materials.

 

Reviewer 2

Comment 1: Thank you for the opportunity to review this revised paper regarding adolescents´ own and assumed parental preferences regarding confidentiality and consent in Belgium. This is an important and well-performed study dealing with a clinically important and difficult issue when treating adolescents. The authors manuscript has improved significantly, and I have only minor comments.

Response: Thank you for your positive feedback on the revised version of our manuscript. Below, you can find a point-by-point response to your minor comments. We hope this version can be accepted for publication in Plos One. 

Comment 2: In the abstract, methods and results are still written in present tense and not past tense as in the main manuscript.

Response: We have revised the tense in the ‘Methods’ and ‘Results’ sections in the abstract.

Background

Comment 3: Line 80-85: you state that evidence from Europe is lacking. However, Thomsen et al have looked at parents’ ambivalent feelings regarding confidentiality in a paper from 2019. The reference from Thomsen, that you do cite is dealing with guidelines for adolescent friendly health services and not only confidentiality and consent. 

Response: Thank you for drawing further attention to this article. We have nuanced our statement regarding the lack of European studies, we now state that European studies are scarce, and highlight Thomsen et al.’s study as one of the exceptions.

Comment 4: Line 96-98: I recommend you to use more recent guidelines regarding time alone during consultations.

Response: We have now also cited two other articles that were already included in our reference list which are more recent (2006 and 2018) that echo the statement in the sentence. 

Comment 5: Line 105-109:Again I recommend newer references e.g.Mcdonagh et al, Acta Paediatrica 2006

Response: Thank you for this suggestion, we have included the reference to McDonagh et al. (2006) and have also included another recent study from Wright et al. (2017) published in Future Healthcare Journal.

Comment 6: Line 113+: the aim of the study should probably be in past tense

Response: We have revised this paragraph and put it in past tense where appropriate. 

Comment 7: Line 123-131: I think that this highly relevant paragraph should be moved up in the background section

Response: Thank you for your thoughtful comments about this paragraph. We certainly agree with you that this paragraph could work higher in the background section, but we struggled to find a good place for it as it balances between methodology and background. In a way, we are arguing how our study participants (a group that we have not yet introduced in the beginning of the background) can be characterised as ‘late adolescents’. We fear that moving it up too high may be confusing for readers. We have moved it above the aim of the study and believe these last two paragraphs now read a little easier.

Materials and methods

Comment 8: Line 139: I guess the students were asked to participate in the study and not in class?

Response: Indeed – we have clarified this sentence.

Comment 9: Line 142-147: I think this should be moved to an Ethics section

I miss a statistics section and suggest that you re-phrase and move the paragraph line 191-200 from the results section

Response: Thank you for these comments. Regarding your comments about the ethics section, Plos One submission guidelines state:

Methods sections of papers on research using human subjects or samples must include ethics statements that specify:

• The name of the approving institutional review board or equivalent committee(s). If approval was not obtained, the authors must provide a detailed statement explaining why it was not needed

• Whether informed consent was written or oral. If informed consent was oral, it must be stated in the manuscript:

o Why written consent could not be obtained

o That the Institutional Review Board (IRB) approved use of oral consent

o How oral consent was documented

Thus, it appears that the statement about ethics must be included in the Methods section, as currently the case.

Regarding the point about line 191-200 (mirroring comments from Reviewer 1), we have created a new header ‘Data analysis’ (just before the Results section, still in the Methods section), which contains the information from the lines you highlighted.

Results

Comment 10: Line 205: please provide a more descriptive headline than T test results. 

Response: We have changed this subtitle to ‘Differences in confidentiality and consent preferences’ and changed the ‘binomial logistic regression’ subtitle to ‘A closer look at late adolescents’ preferences’. 

Comment 11: Line 209-10: You mention SD in the ‘drunk’ case

Response: Following your next comment, we have now removed all numbers in this paragraph.

Comment 12: Line 206-218: This section is a repetition from the table 3 and I think you could omit the numbers and refer to the table. Furthermore, I think that your explanation in your reply to the reviewers regarding how you get the results should be included in the text or table legend as it makes it much more easy to understand.

Response: Following your suggestion, we have now removed the numbers from the text and refer to Table 3 – which has increased the readability of the paragraph. We have also included part of the reply from the comments to the reviewers to the note of Table 3, hoping to clarify some of the confusion regarding the coding of the vignettes.

Comment 13: Line 223: I think it is an overstatement to say that there is limited significant differences as it only applies to the ‘surgery case’ and this could be ‘by chance’. If you think the difference is true, I suggest you discuss this in the discussion section as it is quite interesting.

Response: We agree with your assessment regarding this finding. We have nuanced this finding further, and already discuss these potential gender differences in the discussion (line 286-293). However, we believe more research is needed to draw clear conclusions about such gender differences – our study cannot do so.

Comment 14: 

Line232: we present (present tense) 

Line 236-237: believed (past tense) 

Line 240 supported (past tense)

Response: We have revised all these errors – thank you for noticing them. 

Discussion

Comment 15: Line 253: it seems that there are words missing e.g. ‘perceptions between….’ 

Response: We agree that this sentence ended rather abruptly, and we have extended it somewhat so that it flows better within the text.

Comment 16: Line 262: I would use the reference by Thomsen et al describing parents’ ambivalent views

Response: The study by Thomsen et al. is number 35 in our reference list, so we were already including it in the references for this sentence. Thank you for suggesting it!

Comment 17: Line 271: is there a word missing?

Response: Thank you for noticing this! You are quite right – the word ‘difference’ was missing. We have added it.

Conclusion

Comment 18: Line 322: regardless of parental consent?

Response: We have added ‘parental’ to this sentence.

---

## [Decision Letter · Decision Letter 2]

14 May 2021

PONE-D-20-41061R2

Late Adolescents’ Own and Assumed Parental Preferences Towards Health-Care Related Confidentiality and Consent in Belgium

PLOS ONE

Dear Dr. De Coninck,

Thank you for submitting your manuscript to PLOS ONE. After careful consideration, we feel that it has merit but does not fully meet PLOS ONE’s publication criteria as it currently stands. Therefore, we invite you to submit a revised version of the manuscript that addresses the points raised during the review process.

We look forward to receiving your revised manuscript.

Kind regards,

Prof. Ritesh G. Menezes, M.B.B.S., M.D., Diplomate N.B.

Academic Editor

PLOS ONE

Journal Requirements:

Additional Academic Editor Comments:

• Abstract: Delete the last sentence from the conclusion sub-section.

• Introduction-“Generally, sexual and reproductive health conditions often do not require consent from a parent or guardian.”: Cite a reference.

• Introduction-Lines 52 & 73 (Please note that the line numbers refer to the marked revised manuscript file): Why is ‘’late’’ mentioned within small brackets?

• Introduction-Lines 82-83: Syntax of the sentence needs to be addressed.

• Introduction-Line 83-“Thomson et al. (2019)’’: Cite the reference as per journal style.

• Introduction-Line 99: Cite the reference as per journal style.

• Introduction-Line 125-‘’KU’’: Provide the full form as well.

• Introduction-Lines 113-125: Further add on to mention that ‘’late adolescence” refers to teenage years between the ages of 15 and 19. Cite an appropriate reference to support this statement [possibly, a UNICEF related reference (https://www.unicef.org/sowc2011/pdfs/Early-and-late-adolescence.pdf)].

• Methods-Line 157: It is irrelevant to mention ‘’experiments where humans were involved’’ here. Isn’t it?

• Methods-Lines 160-161: Replace ‘’(sometimes also referred to as ‘cases’ in the rest of this article)’’ with ‘’(cases)’’.

• Methods-Lines 165-167: In addition, specify the related status in Belgium.

• Methods-Line 181: Replace “case’’ with ‘’cases’’.

• Methods-Line 182: Replace ‘’case’’ with ‘’cases’’.

• Methods-Lines 188-192: Would you like to mention the reason for not obtaining information on father’s educational status?

• Results-Line 205: I would prefer to consider using the term ‘’’woman” for an adult female and not an adolescent female. It is not mandatory to revise this accordingly. However, since you have used the term ‘’boys’’ and ‘’girls” in Table 5, it would be better to replace ‘’women’’ with ‘’girls’’.

• Results-Lines 207-208: It is mentioned that over 95% of the sample was between 17 and 19 years of age. Retain this sentence. In addition, mention the age range (17 to 23) in the text and not at the bottom of Table 1. Provide the mean (standard deviation) for age in the text and not in Table 1. In Table 1, provide the number (%) for age 17, age 18, age 19, ages 20-23.

• Results: If I am not mistaken, Table 2 is not referred to in the text.

• Table 2: In the footnote denote the case numbers for drunk/std/pill/surgery.

• Table 3: Indicate the case number in brackets. For example: Drunk (Case 1)

• Table 4: In the footnote denote the case numbers for drunk/std/pill/surgery.

• Table 5: Indicate the case number in brackets. For example: Drunk (Case 1)

• Discussion/Conclusion: Wherever applicable, replace ‘’adolescents’’ with ‘’late adolescents’’. For your information: Early adolescence (10-14 years); Late adolescence (15-19 years). I wish participants 20 and above (age range of participants in the present study: 17-23) were not included in the present study (or their responses were not considered for analysis) in the first place.

• Besides, address the comments made by the reviewer(s).

Reviewers' comments:

Reviewer's Responses to Questions

**Comments to the Author**

1. If the authors have adequately addressed your comments raised in a previous round of review and you feel that this manuscript is now acceptable for publication, you may indicate that here to bypass the “Comments to the Author” section, enter your conflict of interest statement in the “Confidential to Editor” section, and submit your "Accept" recommendation.

Reviewer #1: All comments have been addressed

Reviewer #2: All comments have been addressed

2. Is the manuscript technically sound, and do the data support the conclusions?

Reviewer #1: Yes

Reviewer #2: Yes

3. Has the statistical analysis been performed appropriately and rigorously? 

Reviewer #1: Yes

Reviewer #2: Yes

4. Have the authors made all data underlying the findings in their manuscript fully available?

Reviewer #1: No

Reviewer #2: No

5. Is the manuscript presented in an intelligible fashion and written in standard English?

Reviewer #1: Yes

Reviewer #2: Yes

6. Review Comments to the Author

Reviewer #1: (No Response)

Reviewer #2: Congratulations with the revision of the manuscript. I find it ready for submission after only a few minor comments an suggestions.

The references in the manuscript are not in order starting with 1, 2, 36, 18 etc.

Reference 35 is still the wrong one from Thomsen et al. The correct reference must be: Thomsen EL et al. Int J Adolesc Med Health 2019. doi: 10.1515/ijamh-2018-0226. Online ahead of print.

Line 52: I would erase (late) as I think that health professionals struggle with providing care to adolescents (10-19) and not only late adolescents.

Line 171: I think this sentence is ambiguous as I believe you assess whether adolescents prefer that physicians inform the parents and not whether they think this is within the legal framework.

Line 187: I would erase ‘in the top left and bottom right quadrants as it is difficult to understand if you do not refer to the table. Furthermore, I would erase ‘In what follows’ as this is unnecessary.

Table 5: the lines are difficult to follow

Line 257: I would prefer ‘preference’ instead of perceptions

7. PLOS authors have the option to publish the peer review history of their article (what does this mean?). If published, this will include your full peer review and any attached files.

Reviewer #1: No

Reviewer #2: No

---

## [Author Response · Author response to Decision Letter 2]

17 May 2021

Dear Editor,

Dear Prof. Menezes,

Thank you for your feedback on our paper, which was very helpful. We would also like to thank both reviewers for their positive feedback on the newest version of our manuscript. In this letter, we provide a point-by-point response to each of the remaining comments and suggestions you and Reviewer 2 provided. We hope that this version can be reconsidered for publication in PLOS One.

With warm regards,

David

Editor comments

Comment 1: Abstract: Delete the last sentence from the conclusion sub-section.

Response: Sentence has been removed.

Comment 2: Introduction-“Generally, sexual and reproductive health conditions often do not require consent from a parent or guardian.”: Cite a reference.

Response: We now cite reference at the end of this sentence.

Comment 3: Introduction-Lines 52 & 73 (Please note that the line numbers refer to the marked revised manuscript file): Why is ‘’late’’ mentioned within small brackets?

Response: We removed the small brackets around “late” in line 73, we removed (late) entirely in line 52.

Comment 4: Introduction-Lines 82-83: Syntax of the sentence needs to be addressed.

Response: We have revised this sentence.

Comment 5: Introduction-Line 83-“Thomson et al. (2019)’’: Cite the reference as per journal style.

Response: We have revised this sentence entirely. 

Comment 6: Introduction-Line 99: Cite the reference as per journal style.

Response: Amended.

Comment 7: Introduction-Line 125-‘’KU’’: Provide the full form as well.

Response: There is no full form, the official name of the university is KU Leuven. The KU used to stand for something, but today the institution only uses the abbreviation and never uses the full form.

Comment 8: Introduction-Lines 113-125: Further add on to mention that ‘’late adolescence” refers to teenage years between the ages of 15 and 19. Cite an appropriate reference to support this statement [possibly, a UNICEF related reference (https://www.unicef.org/sowc2011/pdfs/Early-and-late-adolescence.pdf)].

Response: We have added this UNICEF-reference.

Comment 9: Methods-Line 157: It is irrelevant to mention ‘’experiments where humans were involved’’ here. Isn’t it?

Response: But we do mention this, no? Or do you mean to say we should remove this?

Comment 10: Methods-Lines 160-161: Replace ‘’(sometimes also referred to as ‘cases’ in the rest of this article)’’ with ‘’(cases)’’.

Response: Done.

Comment 11: Methods-Lines 165-167: In addition, specify the related status in Belgium.

Response: We have added this info.

Comment 12: Methods-Line 181: Replace “case’’ with ‘’cases’’.

Methods-Line 182: Replace ‘’case’’ with ‘’cases’’.

Response: Done.

Comment 13: Methods-Lines 188-192: Would you like to mention the reason for not obtaining information on father’s educational status?

Response: Previous research has shown that mother’s educational attainment is a robust indicator for socioeconomic status of adolescents. However, given the limited impact of sociodemographic indicators in this analysis, we decided not to spend too many words on this point.

Comment 14: Results-Line 205: I would prefer to consider using the term ‘’’woman” for an adult female and not an adolescent female. It is not mandatory to revise this accordingly. However, since you have used the term ‘’boys’’ and ‘’girls” in Table 5, it would be better to replace ‘’women’’ with ‘’girls’’.

Response: This has been amended.

Comment 15: Results-Lines 207-208: It is mentioned that over 95% of the sample was between 17 and 19 years of age. Retain this sentence. In addition, mention the age range (17 to 23) in the text and not at the bottom of Table 1. Provide the mean (standard deviation) for age in the text and not in Table 1. In Table 1, provide the number (%) for age 17, age 18, age 19, ages 20-23.

Response: We’ve made the necessary changes to table 1 and in the text directly preceding this table.

Comment 16: Results: If I am not mistaken, Table 2 is not referred to in the text.

Response: We did refer to Table 2 in line 209 (now 201).

Comment 17: Table 2: In the footnote denote the case numbers for drunk/std/pill/surgery.

Response: Done.

Comment 18: Table 3: Indicate the case number in brackets. For example: Drunk (Case 1)

Response: Done. 

Comment 19: Table 4: In the footnote denote the case numbers for drunk/std/pill/surgery.

Response: Done.

Comment 20: Table 5: Indicate the case number in brackets. For example: Drunk (Case 1).

Response: Done. 

Comment 21: Discussion/Conclusion: Wherever applicable, replace ‘’adolescents’’ with ‘’late adolescents’’. For your information: Early adolescence (10-14 years); Late adolescence (15-19 years). I wish participants 20 and above (age range of participants in the present study: 17-23) were not included in the present study (or their responses were not considered for analysis) in the first place.

Response: We have gone through both the discussion and conclusion and replaced “adolescents” with “late adolescents” in several instances – a few were left as they were, since they did not refer to adolescents in our study but rather referred to other studies. We appreciate your comments in this regard and will keep this in mind in future studies on the topic.

Comment 22: Besides, address the comments made by the reviewer(s).

Response: All comments by the reviewer have been addressed. Thank you for your useful points.

 

Reviewer 1

No Comments. 

Reviewer 2

Comment 1: Congratulations with the revision of the manuscript. I find it ready for submission after only a few minor comments and suggestions.

Response: We thank you for your positive feedback, and hope that our point-by-point overview of the changes made in accordance with your suggestions is sufficient to accept this paper for publication in Plos One. We thank you for your useful suggestions throughout the review process.

Comment 2: The references in the manuscript are not in order starting with 1, 2, 36, 18 etc.

Response: The order has now been updated.

Comment 3: Reference 35 is still the wrong one from Thomsen et al. The correct reference must be: Thomsen EL et al. Int J Adolesc Med Health 2019. doi: 10.1515/ijamh-2018-0226. Online ahead of print.

Response: We’ve swapped this out with the correct reference now.

Comment 4: Line 52: I would erase (late) as I think that health professionals struggle with providing care to adolescents (10-19) and not only late adolescents.

Response: This has been amended.

Comment 5: Line 171: I think this sentence is ambiguous as I believe you assess whether adolescents prefer that physicians inform the parents and not whether they think this is within the legal framework.

Response: Your assessment is correct, but we do emphasize in this sentence that respondents had to indicate if they were ‘of the opinion’ the physician should inform the parents, which should signal that it was their opinion that was measured instead of their assessment of the legal regulations.

Comment 6: Line 187: I would erase ‘in the top left and bottom right quadrants as it is difficult to understand if you do not refer to the table. Furthermore, I would erase ‘In what follows’ as this is unnecessary.

Response: Both changes have been carried out.

Comment 7: Table 5: the lines are difficult to follow.

Response: We appreciate that this may seem confusing, but we hope that with the copy-editing team of Plos One can make this look much clearer in the published article. If further changes need to be made, we’d be happy to do so.

Comment 8: Line 257: I would prefer ‘preference’ instead of perceptions.

Response: This has been amended.

---

## [Editor Report · Decision Letter 3]

19 May 2021

Late Adolescents’ Own and Assumed Parental Preferences Towards Health-Care Related Confidentiality and Consent in Belgium

PONE-D-20-41061R3

Dear Dr. De Coninck,

We’re pleased to inform you that your manuscript has been judged scientifically suitable for publication and will be formally accepted for publication once it meets all outstanding technical requirements.

Kind regards,

Prof. Ritesh G. Menezes, M.B.B.S., M.D., Diplomate N.B.

Academic Editor

PLOS ONE

---

## [Editor Report · Acceptance letter]

21 May 2021

PONE-D-20-41061R3 

Late Adolescents’ Own and Assumed Parental Preferences Towards Health-Care Related Confidentiality and Consent in Belgium 

Dear Dr. De Coninck:

I'm pleased to inform you that your manuscript has been deemed suitable for publication in PLOS ONE. Congratulations! Your manuscript is now with our production department. 

Kind regards, 

on behalf of

Prof. Dr. Ritesh G. Menezes 

Academic Editor

PLOS ONE